# Spectroelectrochemical analysis of the mechanism of (photo)electrochemical hydrogen evolution at a catalytic interface

Ernest Pastor[1,†], Florian Le Formal[1,†], Matthew T. Mayer[2], S. David Tilley[2,†], Laia Francàs[1], Camilo A. Mesa[1], Michael Grätzel[2] & James R. Durrant[1]

Multi-electron heterogeneous catalysis is a pivotal element in the (photo)electrochemical generation of solar fuels. However, mechanistic studies of these systems are difficult to elucidate by means of electrochemical methods alone. Here we report a spectro-electrochemical analysis of hydrogen evolution on ruthenium oxide employed as an electrocatalyst and as part of a cuprous oxide-based photocathode. We use optical absorbance spectroscopy to quantify the densities of reduced ruthenium oxide species, and correlate these with current densities resulting from proton reduction. This enables us to compare directly the catalytic function of dark and light electrodes. We find that hydrogen evolution is second order in the density of active, doubly reduced species independent of whether these are generated by applied potential or light irradiation. Our observation of a second order rate law allows us to distinguish between the most common reaction paths and propose a mechanism involving the homolytic reductive elimination of hydrogen.

[1] Department of Chemistry, Imperial College London, South Kensington Campus, London SW7 2AZ, UK. [2] Institut des Sciences et Ingénierie Chimiques, Ecole Polytechnique Fédérale de Lausanne (EPFL), Laboratory of Photonics and Interfaces, Station 6, CH-1015 Lausanne, Switzerland. † Present addresses: Molecular Biophysics and Integrated Bioimaging Division, Lawrence Berkeley National Laboratory, One Cyclotron Road, Berkeley, California 94720, USA (E.P.); Institut des Sciences et Ingénierie Chimiques, Ecole Polytechnique Fédérale de Lausanne, Laboratory for Molecular Engineering of Optoelectronic Nanomaterials, Station 6, CH-1015 Lausanne, Switzerland (F.L.F.); Department of Chemistry, University of Zurich, Winterthurerstrasse 190, 8057 Zurich, Switzerland (S.D.T.). Correspondence and requests for materials should be addressed to J.R.D. (email: j.durrant@imperial.ac.uk).

Multi-electron heterogeneous redox catalysis is of increasing importance for the development of sustainable energy systems[1,2]. These include for example the electrolysis or photoelectrolysis of water to release molecular hydrogen, and its utilization for electrical power generation, as well as strategies for $CO_2$ reduction[3–5]. Determination of the mechanisms of such heterogeneous catalytic processes is often challenging. This is particularly the case for multi-electron processes, which may require the accumulation of multiple reduction or oxidizing species to drive the desired transformation[6]. The determination of the rate laws for such reactions in (photo)electrochemical systems, and in particular the dependence of the reaction rate upon the density of reduced/oxidized species, can provide powerful insights into the reaction mechanism. However, such rate law analyses are often difficult to elucidate from conventional electrochemical measurement strategies, except for ideal systems[7,8]. In the study reported herein, we employ a combined optical/electrochemical approach to access and compare the rate law of proton reduction to $H_2$ under conditions of dark and light electrolysis.

Traditionally, precious metals that exhibit near ideal catalytic behaviour such as platinum have been employed as electrocatalysts for the hydrogen evolution reaction (HER)[4,9]. Recently, a new generation of electrocatalysts based on nanostructured metal oxides and sulfides, as well as molecular catalysts, have emerged as potentially lower-cost alternatives[3,10–17]. In this work, we have chosen to focus on a well-established HER electrocatalyst: $RuO_x$ (ref. 18). This electrocatalyst is based upon a nanostructured, amorphous, highly porous $RuO_x$ that herein is deposited onto FTO (fluorine-doped tin oxide) and onto a multilayer $Cu_2O$-based photoelectrode (Fig. 1). Such photocathodes, where the $Cu_2O$ is protected against photocorrosion by thin Al:ZnO (AZO) and $TiO_2$ overlayers, have achieved remarkable solar-to-hydrogen yields and large (near 100%) faradaic efficiencies for solar-driven water splitting[19–21]. This $Cu_2O/AZO/TiO_2/RuO_x$ (referred herein as $[Cu_2O]/RuO_x$) assembly can be considered an example of a 'buried junction' photoelectrode, in which the generation and separation of photogenerated charges in the $Cu_2O/AZO/TiO_2$ layers is at least partially decoupled from the catalytic function by the catalyst overlayer. As such, these systems provide an attractive model for the study of the HER catalysis, as well as enabling a direct comparison of the catalytic function under conditions of dark electrochemical and irradiated photoelectrochemical proton reduction.

In this study, we are concerned with addressing the (photo)-electrocatalytic function of $FTO/RuO_x$ and $[Cu_2O]/RuO_x$. In electrocatalytic systems, the first step toward the determination of a reaction mechanism is the analysis of its current/potential characteristics, often referred to, for metallic electrodes, as a Tafel analysis[7]. Such analyses are typically interpreted in terms of the potential dependence of the free energy offset driving the reaction and can provide information about the nature of the rate-determining step[7]. However, for non-ideal semiconducting or metallic electrodes with multiple redox states, such as the $RuO_x$ studied herein, the interpretation of Tafel analyses is non-trivial and is hampered by the presence of surface and/or intraband states[22]. For photoelectrochemical systems, the interpretation of current/potential data in terms of catalytic function is even more challenging due to the voltage dependence of electron/hole recombination, which can provide a further modulation of flux of charge carriers to the surface as a function of applied potential[23]. As such, for many (photo)electrocatalytic systems, determination of rate laws for multi-redox reactions is a non-trivial challenge, with only limited studies reported in the literature to date.

The key challenge in undertaking purely rate law analyses of heterogeneous redox catalysis is the difficulty in determining the density of reducing species from electrochemical methods alone. In this study, we avoid this limitation by using optical absorbance difference spectroscopy to assay directly the densities of reduced $RuO_x$ species under conditions of both electrochemical and photoelectrochemical proton reduction. This approach opens the possibility to study (photo)electrocatalysis in heterogeneous redox catalysts that are otherwise difficult to study by electrochemical methods alone. Recently, we have shown that it is possible to use such an optical assay to study the photoelectrochemical water oxidation on $\alpha$-$Fe_2O_3$ and $BiVO_4$ photoanodes with high surface areas[24,25]. However, it has not been applied to determine rate laws of electrocatalytic reactions. Employing this approach, we compare the behaviour of $RuO_x$ when it is used as an electrocatalyst and as a part of a $Cu_2O$-based multi-junction photocathode. Taking advantage of the high internal surface area of the $RuO_x$, we demonstrate that we are able to monitor separately the densities of singly and doubly reduced $RuO_x$, and how these densities correlate with the current density for proton reduction. These measurements are used to determine the rate law for both electrochemical and photoelectrochemical proton reduction and thus gain insights into the catalytic mechanism of the HER.

## Results

**Current/potential characteristics.** We start by comparing the current/potential (J-E) characteristics of the $RuO_x$ electrocatalyst in the dark (on FTO) and on the multilayer $[Cu_2O]$ photocathode under irradiation (Fig. 2a). The negative currents observed under reducing bias for both systems are assigned to the

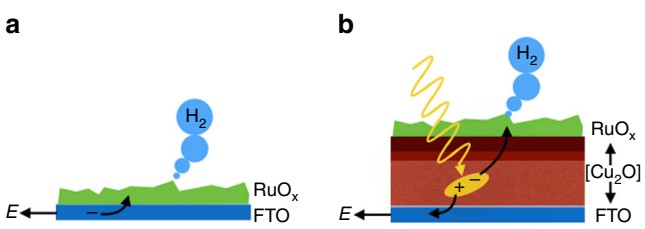

**Figure 1 | Schematic representation of the dark and light electrocatalytic systems.** (**a**) $RuO_x/FTO$ cathode and (**b**) $FTO/Cu_2O/AZO/TiO_2/RuO_x$ photocathode. The diagram is not scaled.

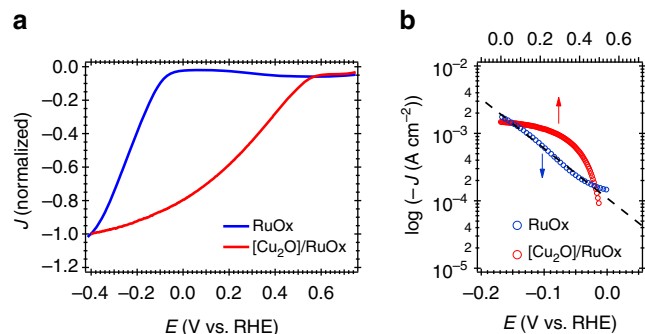

**Figure 2 | J-E behaviour of the dark electrocatalyst and the illuminated photocathode.** (**a**) J-E characteristics normalized for clarity at an applied potential where HER occurs in both systems and (**b**) J-E characteristics represented as log [ − J or − $J^{ph}$] versus E. The samples were measured in a pH = 5 phosphate-sulfate electrolyte.

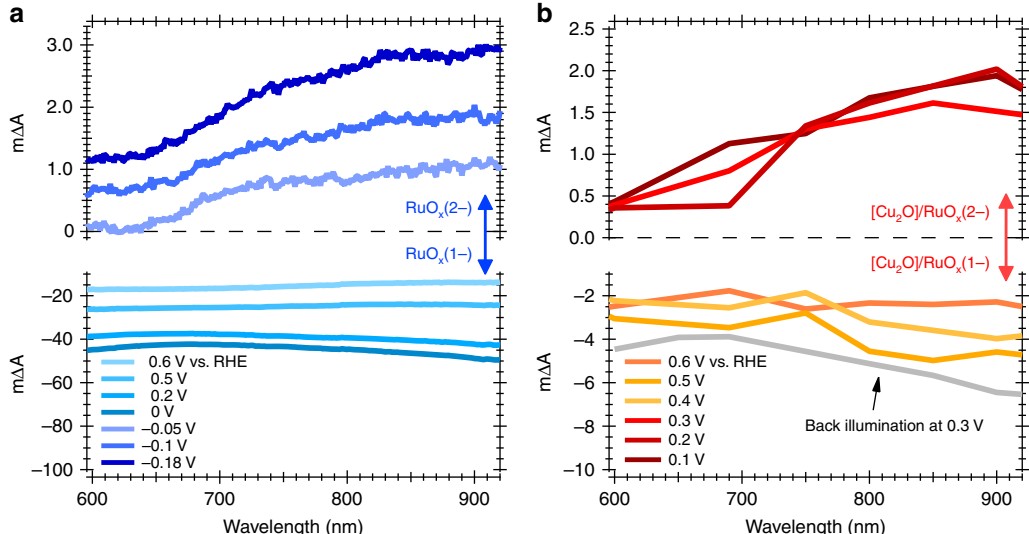

**Figure 3 | Spectroelectrochemical spectra at different applied potentials.** Spectra showing the changes in absorbance ($\Delta A \times 10^{-3}$) when $RuO_x$ (**a**) acts as a dark electrocatalyst and (**b**) is part of a $[Cu_2O]$ photocathode under irradiation. For the lower box in **a**, $\Delta A$ is plotted relative to the absorbance spectrum at open circuit potential (OCP = 0.78 V); for the upper box, showing $\Delta A$ spectra at potentials negative of the hydrogen reduction current onset potential ($E_{onset} = 0$ V versus RHE), the spectra are plotted relative to that at this onset potential. For the $[Cu_2O]/RuO_x$ photocathode (**b**), at all potentials $\Delta A$ is plotted as the difference between light on and light off. For reference, the $\Delta A$ with respect to the absorbance at OCP and the simultaneously measured currents for all applied potentials are shown in Supplementary Figs 3 and 4. For both device types, the negative $\Delta A$ signals observed for $E > E_{onset}$ are assigned to $RuO_x$ reduction to $RuO_x(-1)$, and the positive $\Delta A$ signals observed for $E < E_{onset}$ are assigned to the further reduction to $RuO_x(-2)$ species. For (**b**) the irradiation conditions were ~5-6 s (25 s for back irradiation), 365 nm illumination (~1.5 mW cm$^{-2}$). See Methods for experimental details.

HER deriving from proton reduction[19]. The $J$-$E$ response reveals an approximate 0.5 V difference in the onset of proton reduction catalysis between the electrocatalytic ($E_{onset}$ ~ 0 V versus RHE) and the photocatalytic systems ($E_{onset}$ ~ 0.5 V versus RHE) in agreement with the photovoltage provided by the buried p-$Cu_2O$/n-Al:ZnO junction[26]. It is apparent that the shape of the $J$-$E$ curve of the dark and light-driven systems are qualitatively different despite tracking the same catalytic reaction (HER on $RuO_x$); in particular, the current onset is much sharper for the dark, electrochemically driven reaction. In addition to this, we also observe a small negative current at potentials positive to $E_{onset}$ for both systems. This is also present as a quasi-reversible wave in cyclic voltammogram of $RuO_x$ (Supplementary Fig. 1), and has previously been assigned to the first reduction of the catalyst from $RuO_2$, most probably to $Ru(OH)_3$, (herein referred for simplicity as $RuO_x(1-)$)[27,28].

Figure 2b shows the $J$-$E$ behaviour plotted on a log/linear graph as typically employed for analyses of catalytic function. The dark electrocatalytic current density of $RuO_x$ displays near-linear dependence of log $[-J]$ versus $E$, over the limited potential range shown, with a Tafel slope of 140 mV dec$^{-1}$. As discussed above and elsewhere, the $RuO_x$ studied herein consists of a porous, non-stoichiometric electrocatalytic layer with multiple redox/protonation states, therefore, the direct interpretation of this Tafel slope is complex[29,30]. Moreover, the current density of $[Cu_2O]/RuO_x$ does not show a logarithmic dependence on the applied potential, assigned below to the potential dependence of photoinduced charge recombination. As such, we focus herein on an alternative spectroelectrochemical analysis of these (photo)electrodes.

A key difference in the electrochemical response between electrocatalysts and photoelectrocatalysts is the impact of potential-dependent charge separation or recombination in light-driven systems[23,31]. Such recombination losses can be

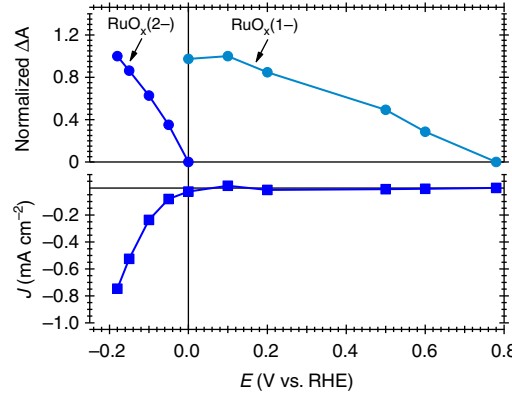

**Figure 4 | Correlation between the changes in absorbance and current.** The $\Delta A$ was calculated as an average of the optical signal between 850 and 900 nm and represented in a normalized scale for clarity. The steady state current density was calculated as the average current measured within 300–350 s (see Supplementary Fig. 4).

observed in the photocurrent ($J^{ph}$) response of the photocathode as sharp cathodic and anodic photocurrent spikes when the light is turned on and off, respectively (Supplementary Fig. 2). Analogous current transients have been reported in other photoelectrodes, and have been assigned to the recombination of charges accumulated at the semiconductor/liquid interface competing with the catalytic reaction[23,32]. The magnitude of these current transients is reduced upon the application of stronger negative potentials, indicating that these partially supress recombination losses. The surface recombination current transients are even more dominant for the $[Cu_2O]$ photocathode alone in the absence of $RuO_x$, indicating that the deposition of

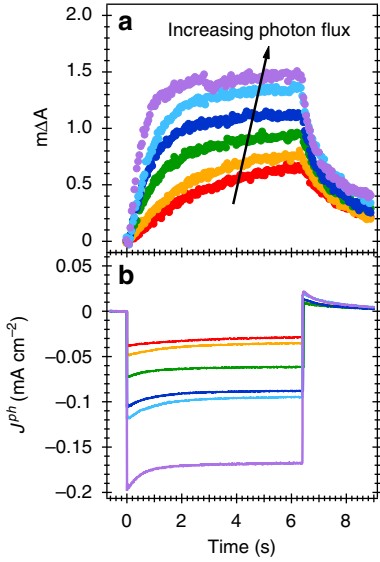

**Figure 5 | Absorbance and photocurrent changes under conditions of HER.** (**a**) Time-dependent photoinduced absorbance changes of the photocathode upon 365 nm light illumination at different photon fluxes (0.5–1.5 mW cm$^{-2}$) at a fixed applied potential of 0.1 V$_{RHE}$ and (**b**) the photocurrent measured simultaneously.

RuO$_x$ results in the partial suppression of these recombination losses. As such, it is apparent that for [Cu$_2$O]/RuO$_x$, the $J$-$E$ behaviour shown in Fig. 2 is primarily determined by the potential dependence of recombination losses, and therefore cannot be used as direct assay of the catalytic behaviour of the RuO$_x$ layer in this system.

**Spectroelectrochemical characteristics.** We turn now to an optical spectroscopy investigation of the RuO$_x$ functioning as a dark HER electrocatalyst on FTO. Figure 3a shows the spectroelectrochemical data for this electrode, plotted as absorbance difference ($\Delta$A) against wavelength at different applied potentials. At potentials negative of the electrode open circuit potential (OCP, 0.78 V$_{RHE}$), a broad negative change in absorbance (bleaching of the absorbance at OCP) is observed across the visible/near-IR region. This loss in absorbance increases with increasing reducing potential until 0 V$_{RHE}$, which corresponds to the onset of the proton reduction electrocatalytic wave, $E_{onset}$. Further increase in applied potential negative of 0 V$_{RHE}$ results in appearance of a new, positive change in absorbance with respect to $E_{onset}$, with increasing amplitude toward the near-IR ($\sim$900 nm). Following our discussion of the electrochemical data above, we assign the negative absorbance at potentials between OCP and $E_{onset}$ to the reduction of RuO$_x$ to its singly reduced species, RuO$_x$(1−), and the positive absorbance at potentials negative of $E_{onset}$ to formation of further reduced catalytic species herein referred as RuO$_x$(2−). Comparison of the amplitudes of these two optical absorbance signals with the steady state catalytic current assigned to the HER suggests that there is correlation between the second reduction of the electrocatalyst to RuO$_x$(2−) and the electrocatalysed HER (Fig. 4). Our optical data indicate that formation of RuO$_x$(2−) is only observed once the first reduction of the electrocatalyst to RuO$_x$(1−) is saturated (see Fig. 4 and also Supplementary Fig. 3). Following literature data, the reduction of RuO$_x$ to RuO$_x$(1−) is most probably associated with Ru$^{IV}$ reduction to Ru$^{III}$ hydroxo species coupled with a lattice expansion[27,28,33], and the second reduction to

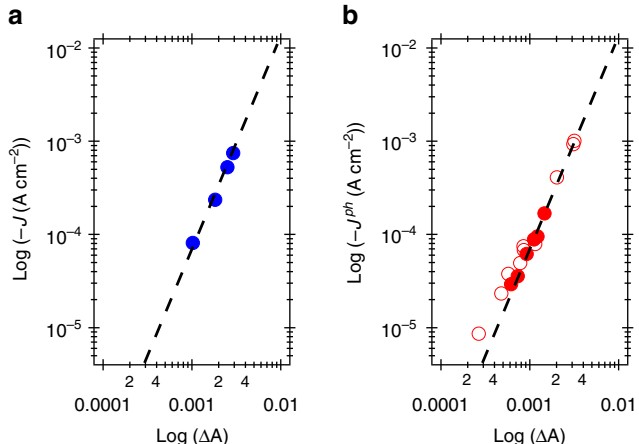

**Figure 6 | Relationship between (photo)current and absorbance changes.** (**a**) Current-absorbance characteristics of the RuO$_x$ electrocatalyst represented as the log [ − $J$] versus the absorbance of the catalytic RuO$_x$ species at different applied potentials (− 0.05 to − 0.2 V versus RHE). (**b**) Photocurrent-absorbance characteristics of the photocathode represented as the log [ − $J^{ph}$] versus the photoinduced absorbance of the catalytic RuO$_x$ species generated at 0.1 V versus RHE upon 365 nm illumination at different photon fluxes ($\sim$0.5 to 1.5 mW cm$^{-2}$). The regression of log [ − $J$] versus log [$\Delta$A] yielded slopes of 2.1 ± 0.1 (correlation coefficient, $R$ = 0.997) for the electrocatalyst. For the photocatalyst two sets of data are shown, collected with (empty circles, see Supplementary Fig. 8) and without (solid circles, see Fig. 5) the addition of 0.01 M surfactant (Tritron X-100) used to promote facile bubble release. The independent data sets have been normalized to be plotted in the same graph. The regression of log [ − $J^{ph}$] versus log [$\Delta$A] of each dataset yielded identical slopes of 1.9 ± 0.1 ($R$ = 0.992).

RuO$_x$(2−) is probably associated with formation of ruthenium hydrides (see 'Discussion' section), although we note that the specific nature of these states is not the primary focus of the study herein.

Next we focus on the spectroelectrochemical analysis of [Cu$_2$O]/RuO$_x$, enabled by its recent adaptation as a transparent photocathode for tandem operation[34]. In this case, current generation, and therefore RuO$_x$ reduction, is both light and voltage dependent. We focus on the change in the absorbance ($\Delta$A) induced by UV (365 nm) irradiation, measured at different applied potentials (Fig. 3b), with control data in the absence of Cu$_2$O showing negligible signals (Supplementary Fig. 5). The resulting data show qualitative similarities to the dark spectroelectrochemical of RuO$_x$ alone (Fig. 3a). At potentials positive of the onset of photocurrent ($>$0.4 V$_{RHE}$), a broad negative change in absorbance is observed; for more negative potentials, where HER photocurrent is monitored, a positive absorbance change is observed. The striking similarity between the absorbance changes induced electrochemically for bare RuO$_x$ and photoelectrochemically for [Cu$_2$O]/RuO$_x$ suggest that in both cases we are primarily monitoring the same species of RuO$_x$(1−) and RuO$_x$(2−). Further evidence that RuO$_x$ dominates the optical signals is found upon back illumination of [Cu$_2$O]/RuO$_x$ under moderate bias (0.3 V$_{RHE}$). Under these conditions in which negligible photocurrent is obtained (Supplementary Fig. 6), only the negative absorbance change assigned to singly reduced, non-catalytically active RuO$_x$(1−) is monitored (Fig. 3b).

We now consider the quantitative correlation between the density of RuO$_x$(2−) species and the HER current. For dark electrocatalysis, this is provided by the optical

and current density data as a function of applied potential (Fig. 4). However, for the photocathode, the potential dependence of recombination losses complicates the interpretation of the equivalent data. We avoid this by measuring the change in absorbance upon light pulses of different intensities at a fixed, moderate reducing potential (0.1 $V_{RHE}$). Figure 5 shows the absorbance change at 900 nm and the associated photocurrent measured as a response to 6 s light pulses. As expected, an increase in the light intensity results in larger steady state photocurrents and larger absorbance changes corresponding to the accumulation of more catalytic $RuO_x(2-)$ species. The decay of the absorbance signal following light off is assigned primarily to the decay of $RuO_x(2-)$ species due to proton reduction, indicating a 0.1s–1s timescale for this reaction (see also Supplementary Fig. 7).

**Rate law analysis**. Figure 6 summarizes our different experimental assays of the catalytic function of $RuO_x$ and $[Cu_2O]/RuO_x$ plotted as the log $[-J]$ (or $-J^{ph}$) against the optical absorbance increase (log $[\Delta A]$) measured at 900 nm, (that is, $RuO_x(2-)$). For $RuO_x$, this log/log representation of the current flow shows a linear dependence with a slope of 2.1 (Fig. 6a). For $[Cu_2O]/RuO_x$, the photocurrent also shows a similar linear dependence on the change in absorbance with a gradient of 1.9 (Fig. 6b). These data provide a clear indication that for both systems, the catalytic current is proportional to approximately the square of the density of $RuO_x(2-)$. Furthermore, the two plots overlay almost exactly, indicating that for both systems, the HER current density depends indistinguishably upon the density of $RuO_x(2-)$. This agreement is particularly striking given the different experimental methods employed—analysis of the dark current as a function of applied bias and of the photocurrent as a function light intensity, and provides confirmation of the validity of the experimental approaches employed herein.

## Discussion

The experimental approach employed herein is based upon the use of an optical assay of the density of doubly reduced $RuO_x$ species ($RuO_x(2-)$) and its correlation with electrocatalytic and photocatalytic proton reduction to hydrogen on FTO/$RuO_x$ and $[Cu_2O]/RuO_x$, respectively. For both systems, this optical absorbance signal is only observed for applied potentials negative of the onset of catalytic HER current; for less reducing potentials, a broad, negative absorbance change is observed, assigned to the single reduction of $RuO_x$ to a pre-catalytic $RuO_x(1-)$ state. Our approach therefore provides a direct way to relate the dependence of the (photo)current to $RuO_x(2-)$. Strikingly, our plots of log$[-J]$ versus log $[\Delta A]$ for electrochemical and photo-electrochemical proton reduction reveal an identical dependence of the current on $RuO_x(2-)$. This observation contrasts with the different $J$-$E$ responses of these systems (Fig. 2). This difference in $J$-$E$ between the electrocatalytic and photocatalytic systems results primarily from the voltage dependence of photoinduced charge separation and recombination in the $[Cu_2O]/RuO_x$ photocathode.

For the dark $RuO_x$ electrocatalyst, the electrochemical $J$-$E$ response shown in Fig. 2b shows a linear relationship between log $[-J]$ and $E$, typical of many electrochemical reactions[22,35,36]. For metals, such behaviour is usually interpreted in terms of differences in reaction overpotential as a function of applied potential. In contrast, our optical assay of $RuO_x(2-)$ reveals that the concentration of this catalytically active species increases approximately linearly with applied

potential above the $E_{onset}$ (see Fig. 4), clearly distinct from ideal metallic or semiconductor behaviour. Such non-ideal behaviour, which may result from multiple reaction intermediates, the protonation of the surface states or other origins of state inhomogeneity[37], makes direct interpretation of electrochemical data alone very challenging. In contrast, our spectroelectrochemical analysis allows us to correlate the observed (photo)electrocatalytic HER current with the density of reduced, catalytic $RuO_x(2-)$, facilitating analysis of $RuO_x$ function. Moreover, this analysis allows us to compare directly the light and dark electrochemistry bypassing the effect of carrier recombination in the $J^{ph}$-$E$ response.

Our analysis of the dark $RuO_x$/FTO and photo-driven $[Cu_2O]/RuO_x$ systems under steady state conditions (Fig. 6) reveals both systems exhibit the same second-order dependence of HER current density upon $RuO_x(2-)$. This observation provides evidence that for this catalyst, the catalytic function is independent of substrate (that is, FTO or $Cu_2O$/AZO/$TiO_2$-buried junction), of the mechanism driving the reaction (that is, electrochemical or photoelectrochemical) and of the means by which the density of $RuO_x(2-)$ is varied (that is, variation of applied potential or light intensity). Furthermore, the analysis indicates that in terms of catalytic function, the $[Cu_2O]/RuO_x$ can indeed be considered a buried junction device, where the catalytic function of the $RuO_x$ is independent of the $[Cu_2O]$ underlayers. However, we note that the $RuO_x$ also appears to have an additional function reducing charge recombination losses in the photocathode (Supplementary Fig. 2), such that function of the $Cu_2O$/AZO/$TiO_2$ junction is dependent upon the presence of the surface catalytic layer.

We consider now the implications of our results for the mechanism of the HER on $RuO_x$. The HER is often described to occur through a homolytic or a heterolytic path involving the formation of metal-hydride intermediates ($[M^{n+1}-H]$)[18,38–40]. In the homolytic mechanism, the $H_2$ evolution step involves the interaction of two hydrides and the reductive elimination of $H_2$ (equation 1). On the other hand, in the heterolytic mechanisms, the hydride intermediate undergoes further reduction and protonation to generate $H_2$ (equation 2).

$$2[M^{n+1}-H] \rightarrow 2M^n + H_2 \qquad (1)$$

$$[M^{n+1}-H] + e^- + H^+ \rightarrow M^n + H_2 \qquad (2)$$

Previous studies of HER on $RuO_2$ have proposed a heterolytic mechanism on the basis of electrochemical Tafel analyses[18,41,42]. However, other literature on electrocatalytic proton reduction on other electrocatalysts has tended to favour homolytic reaction pathways[30,43]. The interpretation of analyses that rely exclusively on $J$-$E$ characteristics is complex for systems with multiple redox intermediates, where the surface behaviour differs from that of a metal, as discussed above[37]. In contrast, our electro-optical analyses of the rate-determining step of the dark electrochemical and light-driven photoelectrochemical HER on $RuO_x$ reveals a second-order reaction with respect to the concentration of $RuO_x(2-)$. Such behaviour is characteristic of bimolecular processes and thus is indicative of a homolytic rather than heterolytic reaction path involving two $RuO_x(2-)$ species.

On the basis of this observation and previous literature, we propose a tentative mechanism in which initially, before $E_{onset}$, $RuO_x$ ($x \leq 2$) undergoes a one electron reduction forming $Ru(OH)_3$ ($RuO_x(1-)$), as previously characterized[27,28,33]. Following this pre-catalytic step, a further reduction of $Ru(OH)_3$ occurs to generate the active species herein referred as $RuO_x(2-)$. Finally, two active species undergo the reductive elimination of $H_2$ regenerating the pre-catalytic state (equation 2). The specific nature of $RuO_x(2-)$ is beyond the

scope of this paper; however, on the basis of the previously suggested mechanisms, we hypothesize this to be a hydride-like species. We note that detailed studies on the nature of the pre-catalytic and catalytic species observed in our system are required to elucidate the mechanism unequivocally. However, our electro-optical analysis provides a powerful tool to discern between the two most common reaction mechanisms and thus to identify a potential mechanistic route.

In conclusion, in this work, we have presented a combined spectroscopic/electrochemical methodology that allows us to assay the densities of redox species directly under conditions of proton reduction and to correlate these with the respective current densities. This methodology thus enables direct access to the rate law for both electrochemical and photoelectrochemical proton reduction, even for non-ideal surfaces with multiple redox states. Our measurements reveal that HER on $RuO_x$ takes place on the 0.1s–1s timescale via a bimolecular mechanism involving two doubly reduced $RuO_x$ species. We find that the $RuO_x$ behaves indistinguishably when employed as an electrocatalyst on FTO or as part of a photoelectrode, with the rate of hydrogen evolution primarily being determined by the density of doubly reduced $RuO_x$ species, independent of whether these species are generated by applied potential or light irradiation. This spectroelectrochemical methodology allows us to simultaneously harness the advantages of electrochemical and optical methods to study the rate law of non-ideal (photo)electrodes. Provided that the catalytic active species can be optically resolved, this approach has the potential to be applicable to other heterogeneous systems. Our studies are ongoing to investigate the wider applicability of our analysis including consideration of other oxide and nitride electrodes and hybrid inorganic/molecular systems.

## Methods

**Photocathode preparation.** The transparent $[Cu_2O]/RuO_x$ photocathode was prepared as reported previously[34]. In summary, electrodes of fluorine-doped tin oxide on glass (FTO, TEC-15, NSG glass) were treated by brief sputter depositions of Au to yield suitable transparent substrates for the electrodeposition of $Cu_2O$, which was performed from a basic solution of lactate-stabilized copper sulfate[20]. A deposition time of 100 min was used, yielding a $Cu_2O$ layer thickness of ~500 nm. This was directly followed by atomic layer deposition of Al:ZnO (20 nm) and $TiO_2$ (100 nm) (Savannah 100, Cambridge Nanotech).

**$RuO_x$ (photo)electrodeposition.** Samples ($Cu_2O$ photocathode and bare FTO) were passivated by opaque epoxy (Loctite Hysol 9461) to define the active area, then were immersed into aqueous solutions of 1.3 mM $KRuO_4$ for galvanostatic electrodeposition of $RuO_x$ (ref. 19). A current density of $-28.3\ \mu A\ cm^{-2}$ was applied to each for 900 s. For the case of the photocathode, the device was illuminated by a solar simulator (1-sun intensity) to facilitate cathodic charge flow. The resulting ~30 nm thick samples were amorphous and highly porous as previously characterized[19,34]. The samples were then ready for electrochemical testing.

**Electrochemical set-up.** All measurements have been performed in a three-electrode photoelectrochemical cell. The cell was filled with ~10 ml of a combined phosphate 0.1-$Na_2SO_4$ 0.4 M electrolyte adjusted at pH = 5 and platinum gauze was used as a counter electrode. The pH was controlled through the experiment with a Hanna HI 83141 pH-meter. The sample was irradiated at the electrolyte/semiconductor interface. Potentials were applied against a silver/silver chloride reference electrode, with saturated KCl solution and converted to potentials against the reversible hydrogen potential ($E_{RHE}$) according to the Nernst equation.

**(Photo)electrochemical set-up.** Photoinduced absorption and transient photocurrent were measured simultaneously as previously described[24]. The samples were irradiated with light pulses produced with the 365 nm LED (5–6 s on/5–6 s off). The probe light source was a tungsten lamp (Bentham IL1 tungsten lamp), and the probe wavelength was selected using a monochromator placed before the sample. Several long-pass and band-pass filters (Comar Instruments) were used to attenuate the pump (LED) light arriving at the detector. Spectroelectrochemistry measurements as a function of applied potential were performed by fitting the photoelectrochemical cell in a Perkin Elmer (Lambda 25) spectrometer. Potentials

were controlled using a PGSTAT101 potentiostat (Metrohom Autolab). The absorbances were measured when the current density attained steady state.

**Data availability.** The data of this study is available upon request.

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

## Acknowledgements

We gratefully acknowledge financial support from the European Research Council (project Intersolar 291482), Swiss National Science Foundation (project: 140709), Swiss Federal Office for Energy (project: PECHouse 3, contract number SI/500090–03). L.F. thanks the EU for a Marie Curie fellowship (658270). E.P. also thanks the EPSRC for a DTP scholarship.

## Author contributions

E.P. and F.L.F. developed the experimental set-up and conducted the experiments. J.R.D. designed the experiments. M.T.M. and S.D.T. prepared the samples. E.P. and J.R.D. wrote the paper, with the help of F.L.F., M.T.M., S.D.T., L.F., C.A.M. and M.G.

## Additional information

**Competing financial interests:** The authors declare no competing financial interests.

