## [Peer Review File · Nature Communications]

Reviewers' comments:

Reviewer #1 (Remarks to the Author):

The authors employ a methodology previously developed to understand electrocatalytic reactions on semiconductor electrodes (e.g. TiO₂) and utilize it to compare the electrochemistry and the photoelectrochemistry of RuO₂ nanoparticles. They find that the reaction mechanism for HER is second order in active site density, and also suggest that the active site is a doubly reduced form of RuO_x. Using the methodology, they are also suggest that both electrochemical and photoelectrochemical HER on RuO₂ progresses through an H-H associative (homolytic) mechanism, rather than a purely heterolytic path, where surface-bound H reacts further with an electron and a proton from the solution to form H₂.

The work is novel, highly interesting and of great importance for furthering the understanding of both electrochemical and photoelectrochemical HER, as well as for the understanding of the catalytic properties of RuO₂. It brings more clarity to the analogies and differences between reactions on metallic and semiconductor oxide electrodes, which is a question of great theoretical and practical interest. Using the optical absorption spectroscopic technique to "count the electrons" on an equal footing in both dark and light electrochemistry is creative and illustrative, and the method seems possible to apply also for other electrode materials and other reactions. It thus seems to have a good potential to influence the field of electrocatalysis in general.

Additionally, the apparent simplicity of the method (utilizing standard electrochemical and spectroscopic instrumentation) is highly appealing. However, the method is limited in that a transparent semiconductor substrate needs to be used, which (as the authors seem to indicate) apparently influences the J-E response in a way that is not relevant for electrodes with metallic conductivity (I return to this point in the discussion below).

Nevertheless, there are a number of questions about the results and the interpretation of the results that need to be raised.

- Structural characterization of the RuO_x coating seems not to have been made. What is the crystal structure of the nanoparticles in the coating? What was the thickness of the layer in both the dark and the photoelectrode case? These details are important both in analyzing the results and for facilitating the possible reproduction of the study.
- It seems reasonable that the particles would assume the rutile RuO₂ structure. However, this leads to a number of new questions. RuO₂ is metallic (has no band gap), as discussed in great detail by e.g. Goodenough (Goodenough, J. B. *Metallic oxides Prog. Solid State Chem.*, 1971, 5, 145–399). This also explains why RuO₂ is black (absorbing visible light). In comparison with TiO₂ (a material examined using the same method presented in the present paper by some of the authors in the early 1990s), which does not absorb light in the visible range, how does this property of RuO₂ affect the

viability of the method presented in the paper?

- A comment on the surface sensitivity of the method would be interesting. It seems that the method would be primarily bulk selective (since the absorbance of the whole electrode coating is measured), and might not give much understanding for reaction species on the surface of the electrode. If this is the case, how does this influence the conclusion regarding RuOx(2-) as the active site for HER?

- A Tafel analysis of the slope for HER on the dark electrode should be carried out. Considering the claims of being able to identify HER as progressing through an H-H associative mechanism, it is valuable to make sure that the kinetics on the electrode actually correspond to those expected on nanoparticulate RuO2. Furthermore, once the Tafel slope has been determined, a more thorough comparison with previous studies can be carried out.

- On page 6, the authors state: "The dark electrocatalytic current density of RuOx displays near-linear dependence of log [-J] vs E, over the limited potential range shown, indicative of rectifying, diode-like behaviour typical of ideal semiconductor-liquid junctions.". However, as I have commented above, RuO2 is not a semiconductor, as the Fermi level is situated in the pi* band of the material. Indeed, their chosen substrate, fluorine-doped SnO2, is a semiconductor, but in since the goal of the analysis is to further the understanding of RuO2, this should be of secondary importance in the analysis.

- Figure 3a and b:

- The wavelength resolution seems to be significantly lower in 3b than in 3a. Why is this?

- Why do the ΔA absorption lines cross in the photocathode case (b), but never in the dark electrocatalyst case (a)?

Additionally, the magnitude of ΔA is much larger in a than in b. Why is that? The similarities between fig 3 a and b, but especially in Fig 6, still support that we are following the same structural alterations in both the dark and the photocatalytic case, but these differences deserve some comment.

- What might be the reason for the sign change in the absorption difference (ΔA)? Why would "RuOx(-1)" cause reduced absorption, and "RuOx(-2)" increased absorption?

- The correlation between HER current density and change in absorbance attributed to formation of RuOx(2-) is highly interesting. However, given that RuOx(-2) would (intuitively) form through reduction of RuOx(-1), why do we not see a gradual crossover between the change in absorbance associated with RuOx(1-) and that associated with RuOx(2-)? Instead, there seems to be a strong discontinuity between ΔA at 0 V and at -0.05 V (also seen in Figure 3).

- Does the discontinuity perhaps indicate that the species in fact might not be singly and double reduced RuOx species, but that they are two completely independent optically absorbing bulk or surface species (or structural alterations)? Figure 4 clearly indicates that only the absorption of the species identified as RuOx(2-) is associated with the HER current. Then, the presentation in the paper might lead the reader astray, as the "RuOx(-1)" specie might not even be associated with the HER mechanism, although it is associated with some alteration of the coating (a possibility would be that it is associated with H migration into the coating, as most recently

discussed in reference 41). Then, the actual process ongoing would be a combination of one structural change, starting even before HER currents are observed, and another (having a stronger effect on the optical absorbance) which is associated with HER. Continuing along that same train of thought, does bubble formation during hydrogen evolution itself cause a change in absorbance? If that is the case, a very pessimistic conclusion would be that the linear increase associated with the HER current in Figure 4 would simply be due to formation of H₂ bubbles, calling the main conclusions of the paper into question! The authors should therefore comment on whether H₂ gas changes the absorbance. Furthermore, a more critical discussion of the possible identity of the structural changes causing the change in optical absorbance is also necessary.

- On page 14, the authors state: ""For metallic electrodes, such behaviour is usually interpreted in terms of differences in reaction overpotential as a function of applied potential. However, the agreement shown in Figure 6 between our electrocatalytic and photocatalytic data demonstrates that reaction overpotential is not the primary determinant of the J-E response." Again, given that RuO₂ is metallic, the semiconductor response observed by the authors either seems not to be relevant, or is only a function of the FTO substrate. Considering these factors, it does not seem warranted to make strong unqualified claims regarding HER on RuO₂ such as "reaction overpotential is not the primary determinant of the J-E response".

- On page 15, the authors state: "Previous studies of HER on RuO₂ have proposed a heterolytic mechanism on the basis of electrochemical, Tafel analyses.^{18,39,40} However, as we have shown the interpretation of these analyses that rely exclusively on J-E characteristics is complex for systems with multiple redox intermediates." I do not fully agree with this description of the work in previous literature. Are the results presented really in conflict with Tafel analyses? Most studies seem not to have been able to find very strong arguments (see e.g. the uncited reference Boodts, J. C. F. & Trasatti, S. Hydrogen evolution on iridium oxide cathodes, *J. Appl. Electrochem.*, 1989, 19, 255–262) for being sure that the final step in the reaction is dependent on just one active site, and theoretical studies account for both heterolytic and homolytic formation mechanisms (Nørskov, J. K.; Bligaard, T.; Logadottir, A.; Kitchin, J. R.; Chen, J. G.; Pandelov, S. & Stimming, U. Trends in the Exchange Current for Hydrogen Evolution *J. Electrochem. Soc.*, The Electrochemical Society, 2005, 152, J23). Furthermore, again, interpreting the behavior of RuO₂ through theory applicable for semiconductors is not reasonable given that RuO₂ is metallic. The overall picture is instead likely that the Tafel analysis gives information about the electrochemical adsorption of H⁺, whereas the method presented in the present paper gives highly valuable complimentary information about the final associative step in the reaction. This is exactly what Boodts and Trasatti (the previously mentioned reference) suggest when writing "The exact nature of Step 4c is not relevant to the kinetic parameters" (right column of page 261 in the referenced paper). The suggestion of the present paper, that the final step is associative, does not invalidate conclusions from Tafel analyses. The paper of Boodts and Trasatti should be included in the discussion and the discussion should be rephrased.

- Finally, there is a small typographical error in reference 40.

Reviewer #2 (Remarks to the Author):

In the manuscript 'Spectroelectrochemical analysis of the mechanism of the (photo)electrochemical hydrogen evolution at a catalytic interface,' Pastor, Formal, Mayer, Tilley, Francas, Mesa, Gratzel and Durrant describe hydrogen evolution catalyzed by RuOx, both in the dark on an FTO electrode and on a multilayer photocathode based on Cu2O. They use a spectroelectrochemical approach to characterize the reaction mechanism, as traditional 'Tafel' analysis of the current/potential characteristics is not suited to this system, as the semiconductors employed are non-ideal and such an analysis would be dominated by things like surface states. By comparing the absorbance associated with the active catalyst to the current density, the authors aim to provide new insight to heterogeneous light-induced proton reduction.

The novelty of this work lies in (1) the use of spectroelectrochemistry to examine the reaction mechanism and (2) the assignment of a bimetallic H2 evolution pathway.

The main finding is that an induced absorption is observed under applied bias (or irradiation + applied bias for the example of the photocathode). This induced absorption is attributed to RuOx(2-), though it is better assigned to the "catalyst resting state." Because it is a 'steady state transient' species, there is nothing to suggest it is simply the RuOx(2-) species, but rather the identity of this species depends on the rate limiting step of catalysis (for instance it could be the hydride intermediate).

The authors correlate the deltaAbs with the current, and find a second order dependence. From this they conclude that H2 proceeds through a bimetallic pathway. Another main finding is that the catalyst functions the same on FTO as on the photocathode.

The study fails to examine the effect of pH on the current and deltaAbs. I think this could be equally telling in trying to diagnose the proton-electron reactivity of this system. In addition, I think it would be interesting to fit the photocurrent decay when the light is switched off. Are second order kinetics observed here?

We thank both reviewers for their positive feedback and comments to our manuscript. In the following we provide a point-by-point response to their questions and direct them to the sections of the manuscript where changes have been made. Please note that in this letter the reviewer's comments appear in bold and italic.

Reviewer #1 (Remarks to the Author):

The authors employ a methodology previously developed to understand electrocatalytic reactions on semiconductor electrodes (e.g. TiO₂) and utilize it to compare the electrochemistry and the photoelectrochemistry of RuO₂ nanoparticles. They find that the reaction mechanism for HER is second order in active site density, and also suggest that the active site is a doubly reduced form of RuO_x. Using the methodology, they are also suggest that both electrochemical and photoelectrochemical HER on RuO₂ progresses through an H-H associative (homolytic) mechanism, rather than a purely heterolytic path, where surface-bound H reacts further with an electron and a proton from the solution to form H₂.

The work is novel, highly interesting and of great importance for furthering the understanding of both electrochemical and photoelectrochemical HER, as well as for the understanding of the catalytic properties of RuO₂. It brings more clarity to the analogies and differences between reactions on metallic and semiconductor oxide electrodes, which is a question of great theoretical and practical interest. Using the optical absorption spectroscopic technique to "count the electrons" on an equal footing in both dark and light electrochemistry is creative and illustrative, and the method seems possible to apply also for other electrode materials and other reactions. It thus seems to have a good potential to influence the field of electrocatalysis in general.

Additionally, the apparent simplicity of the method (utilizing standard electrochemical and spectroscopic instrumentation) is highly appealing. However, the method is limited in that a transparent semiconductor substrate needs to be used, which (as the authors seem to indicate) apparently influences the J-E response in a way that is not relevant for electrodes with metallic conductivity (I return to this point in the discussion below).

Nevertheless, there are a number of questions about the results and the interpretation of the results that need to be raised.

(See below for point-by-point response to the questions)

Authors response to Reviewer #1:

We thank the reviewer for the positive and constructive feedback provided. We have addressed the issues raised and modified the manuscript accordingly. Here we address each point separately.

- 1. Structural characterization of the RuO_x coating seems not to have been made. What is the crystal structure of the nanoparticles in the coating? What was the thickness of the layer in both the dark and the photoelectrode case? These details are important both in analyzing the results and for facilitating the possible reproduction of the study.***

The synthesis and characteristics of the photocathode and the RuO_x catalytic layer are described in detail in references: *Adv. Funct. Mater.* 2014, 3, 303 (ref: 19) and *Adv. Energy Matter.* 2015, 5, 1501537 (ref: 34). The as deposited RuO_x consisted of a non-stoichiometric amorphous layer, approximately 30 nm thick, which did not produce any detectible crystalline phase by XRD examination. We have updated this information in the Methods section.

- 2. It seems reasonable that the particles would assume the rutile RuO_2 structure. However, this leads to a number of new questions. RuO_2 is metallic (has no band gap), as discussed in great detail by e.g. Goodenough (Goodenough, J. B. *Metallic oxides Prog. Solid State Chem.*, 1971, 5, 145–399). This also explains why RuO_2 is black (absorbing visible light). In comparison with TiO_2 (a material examined using the same method presented in the present paper by some of the authors in the early 1990s), which does not absorb light in the visible range, how does this property of RuO_2 affect the viability of the method presented in the paper?**

These electrodeposited RuO_2 films produce no detectable evidence by XRD that the rutile phase is formed, and given the thickness analysis in the *Adv. Funct. Mater.* study (ref: 19) we believe that the as-deposited RuO_2 is amorphous. The deposition of the thin layer of RuO_x indeed resulted in a slight darkening of the surface and a small loss of the solar incident power conversion (*Adv. Funct. Mater.* 2014, 3, 303) however, the photocathode remained sufficiently transparent (specially under applied potential) to perform measurements in transmission mode. Furthermore, we note that due to its non-stoichiometric, porous, amorphous nature, our catalyst cannot be described as purely metallic, thus the lack of strong black coloration typical of metallic RuO_2 . This is supported by the observation of multiple redox states at different applied potentials (see Figure 4 and response to comment #9 and 10). Text addressing this point has been added on page 6 (lines: 1-3).

- 3. A comment on the surface sensitivity of the method would be interesting. It seems that the method would be primarily bulk selective (since the absorbance of the whole electrode coating is measured), and might not give much understanding for reaction species on the surface of the electrode. If this is the case, how does this influence the conclusion regarding $\text{RuO}_x(2-)$ as the active site for HER?**

The reviewer is right to point this out. However, we note that the RuO_x studied herein is a highly porous film rather than a dense and crystalline layer. Therefore we expect a much higher

internal surface area over the 30 nm layer and surface events to dominate the ΔA signals. As suggested we have added further details about the sample on page 6 (lines: 1-3) and page 17 (lines: 20-21). However, we note that the detailed structure, and internal surface area, of the RuO_x layer is not well characterised, so we prefer not to make strong comment on this point in the manuscript.

4. A Tafel analysis of the slope for HER on the dark electrode should be carried out. Considering the claims of being able to identify HER as progressing through an H-H associative mechanism, it is valuable to make sure that the kinetics on the electrode actually correspond to those expected on nanoparticulate RuO_2 . Furthermore, once the Tafel slope has been determined, a more thorough comparison with previous studies can be carried out.

As suggested we have included a Tafel analysis of the J - E characteristics shown in Figure 2. The analysis yields a slope of 140 mV dec^{-1} . This value differs from those measured for metallic RuO_2 which are typically in the $40\text{-}60 \text{ mV dec}^{-1}$ range. We attribute this discrepancy to the amorphous, non-stoichiometric nature of our catalyst. These surface properties significantly complicate the elucidation of surface coverage and the interpretation of the Tafel slope. A thorough comparison of the Tafel slope of RuO_x with those in the literature is hampered not only by the different electrochemical conditions (herein dictated by the photocathode structure) but also by the difference in the nature of the catalyst. We note that the primary goal of the present paper is to compare dark and light electrochemistry and, as shown in Figure 2, direct comparison of the J - E response for these systems is not possible. For these reasons we believe that a detailed study of the Tafel slope is beyond the scope of the paper. Nonetheless, we have reconsidered our discussion section to address the reviewers concerns on this matter and clarified the text (see also response to comment #11 and 12). Specifically we have revised the text on page: 6 (lines: 1-3 and 5-6), page 13 (lines: 20-25) and page 14 (lines: 1-6).

5. ***On page 6, the authors state: "The dark electrocatalytic current density of RuO_x displays near-linear dependence of log [-J] vs E, over the limited potential range shown, indicative of rectifying, diode-like behaviour typical of ideal semiconductor-liquid junctions.". However, as I have commented above, RuO₂ is not a semiconductor, as the Fermi level is situated in the pi* band of the material. Indeed, their chosen substrate, fluorine-doped SnO₂, is a semiconductor, but in since the goal of the analysis is to further the understanding of RuO₂, this should be of secondary importance in the analysis.***

We thank the reviewer for the remarks on this aspect and have changed the text throughout to avoid any suggestion that the RuO_x is a semiconductor. This includes the remarks on ideal semiconductor behaviour made in the introduction (page 3,4) and in the discussion (see response to comment #11 and 12). Specifically, the statement quoted in the referee's comment #5 now on page 5 (lines: 23-25) and page 6 (lines: 1-6) reads as follows:

"The dark electrocatalytic current density of RuO_x displays near-linear dependence of log [-J] vs E, over the limited potential range shown, with a Tafel slope of 140 mV dec⁻¹. As discussed above and elsewhere, the RuO_x studied herein consists of a porous, non-stoichiometric electrocatalytic layer with multiple redox / protonation states, therefore the direct interpretation of this Tafel slope is complex.^{29,30} Moreover, the current density of [Cu₂O]/RuO_x does not show a logarithmic dependence on the applied potential, assigned below to the potential dependence of photoinduced charge recombination. As such we focus herein on an alternative spectroelectrochemical analysis of these (photo)electrodes."

6. ***Figure 3a and b: The wavelength resolution seems to be significantly lower in 3b than in 3a. Why is this?***
7. ***Why do the ΔA absorption lines cross in the photocathode case (b), but never in the dark electrocatalyst case (a)? Additionally, the magnitude of ΔA***

is much larger in a than in b. Why is that? The similarities between fig 3 a and b, but especially in Fig 6, still support that we are following the same structural alterations in both the dark and the photocatalytic case, but these differences deserve some comment.

As points 6 and 7 refer to the measurement of the absorption changes we address them together:

The changes in absorbance measured shown in Figure 3a and 4b were collected in different instruments. A *PerkinElmer* spectrometer was used for spectroelectrochemical measurements whereas a home-built Transient Absorption Spectrometer, that allowed simultaneous illumination of the sample, was used for the light induced measurements. The way the spectral scans are performed and the wavelength resolution is different between the two instruments. This accounts for the differences observed in the figure. Nonetheless, as the reviewer points out, the similarities between the two spectra suggest that they represent the same changes. The correlation in Figure 6, which is an independent measurement collected at a fixed wavelength as a function of the photon flux (rather than scanning the energy, see Figure 5) further supports this assignment.

We have not included experimental information in the figure caption due to limits in the text length. Nonetheless, for clarity on the experimental differences we have added a note at the end of the caption (page 9) to point the reader to the methods section where these details are given.

8. What might be the reason for the sign change in the absorption difference (ΔA)? Why would " $\text{RuO}_x(-1)$ " cause reduced absorption, and " $\text{RuO}_x(-2)$ " increased absorption?

Both in the dark and light-driven electrocatalytic systems we observe that $\text{RuO}_x(1-)$ absorbs less than the starting RuO_x species (at OCP = 0.78 V_{RHE}) and that $\text{RuO}_x(2-)$ absorbs more than

RuO_x(1-). The absorption spectra of Ru complexes is well known to be strongly modulated by the Ru oxidation state, although the details depend on the first coordination sphere of the ruthenium atom. As we have not performed a detailed study on the nature of the RuO_x species, it is difficult for us to comment directly on their expected absorbance changes.

- 9. *The correlation between HER current density and change in absorbance attributed to formation of RuO_x(2-) is highly interesting. However, given that RuO_x(-2) would (intuitively) form through reduction of RuO_x(-1), why do we not see a gradual crossover between the change in absorbance associated with RuO_x(1-) and that associated with RuO_x(2-)? Instead, there seems to be a strong discontinuity between ΔA at 0 V and at -0.05 V (also seen in Figure 3).***
- 10. *Does the discontinuity perhaps indicate that the species in fact might not be singly and double reduced RuO_x species, but that they are two completely independent optically absorbing bulk or surface species (or structural alterations)? Figure 4 clearly indicates that only the absorption of the species identified as RuO_x(2-) is associated with the HER current. Then, the presentation in the paper might lead the reader astray, as the "RuO_x(-1)" specie might not even be associated with the HER mechanism, although it is associated with some alteration of the coating (a possibility would be that it is associated with H migration into the coating, as most recently discussed in reference 41). Then, the actual process ongoing would be a combination of one structural change, starting even before HER currents are observed, and another (having a stronger effect on the optical absorbance) which is associated with HER. Continuing along that same train of thought, does bubble formation during hydrogen evolution itself cause a change in absorbance? If that is the case, a very pessimistic conclusion would be that the linear increase associated with the HER current in Figure 4***

would simply be due to formation of H₂ bubbles, calling the main conclusions of the paper into question! The authors should therefore comment on whether H₂ gas changes the absorbance. Furthermore, a more critical discussion of the possible identity of the structural changes causing the change in optical absorbance is also necessary.

As comments 9 and 10 refer to the identity of the observed species we address them together. Regarding the 'discontinuity', we suspect the reviewer may have misunderstood how these spectra were calculated. There is no discontinuity in the absolute absorbance between 0 and -0.05 V, only a change in the direction of change of absorbance with voltage (as shown in Figure S3), with the initial negative absorption change saturating, and a new, spectrally shifted positive absorbance change appearing. The data for -0.05 V and more negative is therefore plotted relative to the 0.0 V spectrum. This behavior, observed in both the light and dark electrocatalytic system, suggests that the formation of RuO_x(2-) species is only observed once the first reduction of RuO_x to RuO_x(1-) takes place. Such behaviour further supports our assignment to singly reduced (pre-catalytic) and doubly reduced (catalytic) species. We hypothesise that the formation of RuO_x(1-) is associated with the reduction of Ru^{IV} to Ru^{III} hydroxo species and the reduction to RuO_x(2-) involves the formation of a hydride. As the reviewer points out, these processes might be associated with structural changes. We have made the following modification to the text to reflect these points:

On page 8 (lines: 2-9):

“Our optical data indicate that formation of RuO_x(2-) is only observed once the first reduction of the electrocatalyst to RuO_x(1-) is saturated (see Figure 4 and also Figure S3). Following literature data, the reduction of RuO_x to RuO_x(1-) is most probably associated with Ru^{IV} reduction to Ru^{III} hydroxo species coupled with a lattice expansion,^{27,28,33} and the second reduction to RuO_x(2-) is probably associated with formation of ruthenium hydrides (see discussion section), although we note that the specific nature of these states is not the primary focus of the study herein.”

On page 16 (lines: 3-4)

“We note that detailed studies on the nature of the pre-catalytic and catalytic species observed in our system are required to unequivocally elucidate the mechanism.”

Regarding the possible contribution of bubble formation to the absorption change in the visible region, we note that large bubble formation was observed at high current density. This resulted in noise in our optical signal but no net changes in amplitude. This effect limited our dataset to the current densities described in the manuscript. Additionally, in Figure 6b we showed that the addition of surfactant to decrease bubble formation resulted in essentially identical response, further verifying that the bubbles did not play the dominant role in the spectral observations.

11. ***On page 14, the authors state: “For metallic electrodes, such behaviour is usually interpreted in terms of differences in reaction overpotential as a function of applied potential. However, the agreement shown in Figure 6 between our electrocatalytic and photocatalytic data demonstrates that reaction overpotential is not the primary determinant of the J-E response.” Again, given that RuO₂ is metallic, the semiconductor response observed by the authors either seems not to be relevant, or is only a function of the FTO substrate. Considering these factors, it does not seem warranted to make strong unqualified claims regarding HER on RuO₂ such as “reaction overpotential is not the primary determinant of the J-E response”.***
12. ***On page 15, the authors state: “Previous studies of HER on RuO₂ have proposed a heterolytic mechanism on the basis of electrochemical, Tafel analyses.^{18,39,40}***

However, as we have shown the interpretation of these analyses that rely exclusively on J-E characteristics is complex for systems with multiple redox intermediates." I do not fully agree with this description of the work in previous literature. Are the results presented really in conflict with Tafel analyses? Most studies seem not to have been able to find very strong arguments (see e.g. the uncited reference Boodts, J. C. F. & Trasatti, S. Hydrogen evolution on iridium oxide cathodes, J. Appl. Electrochem., 1989, 19, 255–262) for being sure that the final step in the reaction is dependent on just one active site, and theoretical studies account for both heterolytic and homolytic formation mechanisms (Nørskov, J. K.; Bligaard, T.; Logadottir, A.; Kitchin, J. R.; Chen, J. G.; Pandelov, S. & Stimming, U. Trends in the Exchange Current for Hydrogen Evolution J. Electrochem. Soc., The Electrochemical Society, 2005, 152, J23). Furthermore, again, interpreting the behavior of RuO₂ through theory applicable for semiconductors is not reasonable given that RuO₂ is metallic. The overall picture is instead likely that the Tafel analysis gives information about the electrochemical adsorption of H⁺, whereas the method presented in the present paper gives highly valuable complementary information about the final associative step in the reaction. This is exactly what Boodts and Trasatti (the previously mentioned reference) suggest when writing "The exact nature of Step 4c is not relevant to the kinetic parameters" (right column of page 261 in the referenced paper). The suggestion of the present paper, that the final step is associative, does not invalidate conclusions from Tafel analyses. The paper of Boodts and Trasatti should be included in the discussion and the discussion should be rephrased.

As points 11 and 12 refer to the electrochemical analysis and the reaction mechanism we address them together:

We thank the reviewer for the detailed suggestions regarding the reaction mechanism and the analysis of the J - E response of RuO_x . We agree that our electro-optical analysis does not invalidate previous conclusions based on Tafel analysis and that indeed the information provided by each method might be complementary. Nonetheless we highlight that the interpretation of J - E characteristics is complex for non-ideal metallic surfaces as it is the case of our RuO_x and cannot be applied for light-driven systems. In such cases our electro-optical analysis provides a direct way to access information about the rate determining step. We have clarified this in the text. Specifically, the statements quoted in the referee's comment #11 and 12 now read as follows:

On page 13 (lines: 17-25) and page 14 (lines: 1-6)

“For the dark RuO_x electrocatalyst, the electrochemical J - E response shown in Figure 2b shows a linear relationship between $\log [-J]$ and E , typical of many electrochemical reactions.^{22,35,36} For metals, such behaviour is usually interpreted in terms of differences in reaction overpotential as a function of applied potential. In contrast, our optical assay of $\text{RuO}(2-)$ reveals that the concentration of this catalytically active species increases approximately linearly with applied potential above the E_{onset} (see Figure 4), clearly distinct from ideal metallic or semiconductor behaviour. Such non-ideal behaviour, which may result from multiple reaction intermediates, the protonation of the surface states or other origins of state inhomogeneity,³⁷ makes direct interpretation of electrochemical data alone very challenging. In contrast our spectroelectrochemical analysis allows us to correlate the observed (photo)electrocatalytic HER current with the density of reduced, catalytic $\text{RuO}_x(2-)$, facilitating analysis of RuO_x function. Moreover, this analysis allows us to directly compare light and dark electrochemistry bypassing the effect of carrier recombination in the J^{ph} - E response.”

On page 15 (lines: 8-18)

“Previous studies of HER on RuO_2 have proposed a heterolytic mechanism on the basis of electrochemical Tafel analyses.^{18,41,42} However, other literature on electrocatalytic proton reduction on

other electrocatalysts has tended to favour homolytic reaction pathways.^{30,43} The interpretation of analyses that rely exclusively on J-E characteristics is complex for systems with multiple redox intermediates, where the surface behaviour differs from that of a metal, as discussed above.³⁷ In contrast our electro-optical analyses of the rate determining step of the dark electrochemical and light driven photoelectrochemical HER on RuO_x reveals a second order reaction with respect to the concentration of RuO_x(2-). Such behaviour is characteristic of bimolecular processes and thus is indicative of a homolytic rather than heterolytic reaction path involving two RuO_x(2-) species.”

13. Finally, there is a small typographical error in reference 40.

The typographical error has been corrected, please note that this reference is now reference 42.

Reviewer #2 (Remarks to the Author):

In the manuscript ‘Spectroelectrochemical analysis of the mechanism of the (photo)electrochemical hydrogen evolution at a catalytic interface,’ Pastor, Formal, Mayer, Tilley, Francas, Mesa, Gratzel and Durrant describe hydrogen evolution catalyzed by RuOx, both in the dark on an FTO electrode and on a multilayer photocathode based on Cu₂O. They use a spectroelectrochemical approach to characterize the reaction mechanism, as traditional ‘Tafel’ analysis of the current/potential characteristics is not suited to this system, as the semiconductors employed are non-ideal and such an analysis would be dominated by things like surface states. By comparing the absorbance associated with the active catalyst to the current density, the authors aim to provide new insight to heterogeneous light-induced proton reduction.

The novelty of this work lies in (1) the use of spectroelectrochemistry to examine the reaction mechanism and (2) the assignment of a bimetallic H₂ evolution pathway.

The main finding is that an induced absorption is observed under applied bias (or irradiation + applied bias for the example of the photocathode). This induced absorption is attributed to RuOx(2-), though it is better assigned to the “catalyst resting state.” Because it is a ‘steady state transient’ species, there is nothing to suggest it is simply the RuOx(2-) species, but rather the identity of this species depends on the rate limiting step of catalysis (for instance it could be the hydride intermediate).

The authors correlate the deltaAbs with the current, and find a second order dependence. From this they conclude that H₂ proceeds through a bimetallic pathway. Another main finding is that the catalyst functions the same on FTO as on the photocathode.

The study fails to examine the effect of pH on the current and deltaAbs. I think this could be equally telling in trying to diagnose the proton-electron reactivity of this system. In

addition, I think it would be interesting to fit the photocurrent decay when the light is switched off. Are second order kinetics observed here?

Authors response to Reviewer #2:

We thank the reviewer for the positive comments and the feedback. With regards to identity of the RuO_x species we have clarified our hypothesis on the text; we have added the following statement on page 8 (lines 4-9):

“Following literature data, the reduction of RuO_x to $\text{RuO}_x(1-)$ is most probably associated with Ru^{IV} reduction to Ru^{III} hydroxo species coupled with a lattice expansion,^{27,28,33} and the second reduction to $\text{RuO}_x(2-)$ is probably associated with formation of ruthenium hydrides (see discussion section), although we note that the specific nature of these states is not the primary focus of the study herein.”

Regarding the potential study on the effect of the pH, we agree that this could be a very informative study and in fact similar work is currently ongoing in our group. However, the main goal of the present manuscript is to compare the dark and light electrochemistry and in this case the operation conditions were dictated by the optimum stable operating conditions of the photocathode.

Regarding the photocurrent transients, we note that in the multilayer photocathode these transients are primarily sensitive to transport properties in the absorbing layer (Cu_2O) and therefore their interpretation is not straightforward. On the other hand, we have assigned the decay of the photoinduced absorbance when the light is turned off (see Figure 5) to primarily the decay of the $\text{RuO}_x(2-)$ species due to proton reduction; this suggests that this reaction is

occurring in the 100 ms-s timescale following second order kinetics. We have added a figure of this optical signal decays in the Supplementary Information (Figure S8).

REVIEWERS' COMMENTS:

Reviewer #1 (Remarks to the Author):

The authors have now answered my comments and revised the paper taking all my suggestions into account. I support the publication of the paper in its current form (I did notice one very minor spelling error in the word "spectroelectrochemical" on line 250 of page 14).

Reviewer #2 (Remarks to the Author):

In the revised manuscript 'Spectroelectrochemical analysis of the mechanism of the (photo)electrochemical hydrogen evolution at a catalytic interface,' Pastor, Formal, Mayer, Tilley, Francas, Mesa, Gratzel and Durrant have addressed the concerns I raised in my initial review. I find the work satisfactory for publication.